# Babassu Coconut Fibers: Investigation of Chemical and Surface Properties (*Attalea speciosa*.)

**DOI:** 10.3390/polym15193863

**Published:** 2023-09-23

**Authors:** Yago Soares Chaves, Pedro Henrique Poubel Mendonça da Silveira, Sergio Neves Monteiro, Lucio Fabio Cassiano Nascimento

**Affiliations:** 1Department of Materials Science, Military Institute of Engineering-IME, Praça General Tíburcio, 80, Urca, Rio de Janeiro 222290-270, RJ, Brazil; snevesmonteiro@gmail.com (S.N.M.); lucio@ime.eb.br (L.F.C.N.); 2West Zone Campus, Rio de Janeiro State University–UERJ, Avenida Manuel Caldeira de Alvarenga, 1203, Campo Grande, Rio de Janeiro 23070-200, RJ, Brazil; pedroo.poubel@gmail.com

**Keywords:** babassu fiber, diameter, chemical properties, X-ray diffraction properties, surface morphological

## Abstract

To complement previous results, an analysis of the chemical and morphological properties of babassu fibers (*Attalea speciosa Mart. ex Spreng*.) was conducted in order to evaluate their potential as reinforcements in the production of composites with epoxy matrix. The diameter distribution was analyzed in a sample of one hundred fibers, allowing the verification of its variation. The determination of the chemical properties involved experimental analyses of the constituent index and X-ray diffraction. The diffractogram was used to calculate the crystallinity index and the microfibril angle, which are crucial parameters that indicate the consistency of the mechanical properties of babassu fibers and the feasibility of their use in composites. The results revealed that babassu fiber has a chemical composition, with contents of 28.53% lignin, 32.34% hemicellulose, and 37.97% cellulose. In addition, it showed a high crystallinity index of 81.06% and a microfibril angle of 7.67°. These characteristics, together with previous results, indicate that babassu fibers have favorable chemical and morphological properties to be used as reinforcements in composites, highlighting its potential as an important material for applications in technology areas.

## 1. Introduction

Seeking to contribute to sustainable development and meet economic demands, researchers are investigating alternative materials as substitutes for traditional ones in polymeric composites. Among the various materials under scrutiny, composites reinforced with natural fibers have garnered attention for their exceptional ballistic properties and favorable performance across multiple applications [1,2,3,4].

Composites reinforced with natural fibers have sparked considerable interest in both the scientific community and industry. This interest primarily stems from the advantages they offer over traditional composites, such as reduced weight, enhanced mechanical strength, diminished environmental impact, and lower cost [5,6,7].

Brazil possesses the potential to develop materials utilizing natural fibers abundantly available within its territory, including jute, sisal, coconut, and buriti among others. These fibers serve as reinforcements for polymeric matrices, imparting excellent mechanical properties suitable for applications in sectors like construction, automotive, and aerospace. By promoting the use of these reinforced composites, local economies can flourish while also contributing to sustainability by reducing reliance on non-renewable resources. Investments in research in this field are crucial for exploring the potential of these materials, fostering sustainable development, and driving technological innovation [4,8,9,10].

Lignocellulosic natural fibers possess heterogeneous characteristics with varying mechanical properties among the fibers. The growth process of these fibers is regulated by the metabolism of plant cells. Several factors influence their growth, including soil composition, water availability, light exposure, and genetic variation [11]. Reducing the fiber diameter tends to enhance the mechanical properties as it reduces internal defects, resulting in finer fibers with improved mechanical properties [12,13].

The fibers obtained from the fruit of the babassu palm, illustrated in Figure 1a, belong to the Arecaceae family, specifically the Attalea genus. In Brazil, the original species is scientifically known as *Attalea speciosa*. This palm tree is native to a biome called the coconut palm forest, which spans from the Amazon to the caatinga regions, with a greater presence in the states of Maranhão and Piauí [14,15]. The babassu palm can grow up to 20 m tall, and its fruits contain edible oil seeds, as per the illustration in Figure 1b. Each palm tree typically produces an average of 150 to 250 coconuts per bunch, with four bunches per palm. The fruits have an ellipsoidal shape, measuring 8 to 15 cm in length and 5 to 7 cm in diameter [16].

Babassu exhibits various applications, as exemplified by the study conducted by Silva Lima et al. [17], which explored the production of activated carbon as an advanced sustainable carbon material. Bauer et al. [18] examined the physical and chemical characteristics of babassu oil for its potential use in the pharmaceutical and biofuel industries. Zanine et al. [19] proposed a straightforward solution for feeding dairy cows by analyzing certain chemical properties. Moura et al. [20] demonstrated the production of thermoplastic starch composites derived from starch and babassu fibers. Despite these diverse applications, babassu fiber remains relatively underexplored as a material for engineering purposes.

In a recent article [21], the preliminary results on babassu fibers were investigated for their potential engineering applications as reinforcements for polymer matrix composites. Table 1 illustrates the previous results obtained on the mechanical properties and density of babassu fibers with different diameters.

The article [21] not only explored the thermal properties of babassu fibers but also began a comprehensive chemical investigation. The study focused particularly on the degradation of the fiber in response to increasing temperature as well as on the analysis of the functional groups intrinsic to the fiber structure. Figure 2 shows the graphs resulting from the differential scanning calorimetry (DSC) and thermogravimetry (TGA) analyses, while Figure 3 illustrates the spectrum obtained using Fourier transform infrared spectroscopy (FTIR).

According to the article [21], the degradation of babassu fiber, as observed in the TGA graphs, can be divided into different stages. The first stage of degradation begins at a temperature of 37.8 °C and extends to 200 °C, resulting in a 13.6% reduction in the mass of the babassu fiber. The starting temperature for the thermal degradation of babassu fiber is 251.1 °C. The second stage of degradation occurs between 200 °C and 382 °C and is characterized by the greatest loss of mass, with 69.9% of the total mass of babassu fiber lost at a maximum temperature of 346.2 °C. The third, and final, stage of babassu fiber degradation begins at 382 °C and continues up to 564 °C, culminating in the complete degradation of the fiber.

The DTG curves show the peaks referring to the degradation temperatures of the babassu fiber, with the degradation process of the babassu fibers initially occurring at 37.8 °C. The second peak was observed at 287.8 °C, followed by the third peak at 346.2 °C. The last degradation peak, present in the third stage of fiber degradation, is located at 557.5 °C. This peak indicates the final degradation of the babassu fiber and its complete transformation into ash after the test.

In the differential scanning calorimetry (DSC) analysis graph, we can identify several thermal transitions. Firstly, we observe an endothermic peak at 67.8 °C, indicating the evaporation of water from the fiber. Next, we identified three distinct exothermic peaks at temperatures of 132.1, 406, and 478.5 °C. The peak at 132.1 °C is probably associated with the decomposition of lignin and part of the degradation of cellulose I. The peak at 406 °C is related to the degradation of hemicellulose components and the decomposition of α-cellulose. Finally, the third peak at 478.5 °C corresponds to the final degradation of lignin and cellulose.

In the Fourier transform infrared spectroscopy (FTIR) spectrum, we observed several significant bands. The most prominent band occurs at 3286 cm^−1^ and is interpreted as a stretch in the (OH) bond. The adjacent band at 2919 cm^−1^ is associated with the characteristics of the macromolecules that make up the fiber. The band located at 1613 cm^−1^ can be attributed to the functional groups belonging to hemicellulose. The band at 1317 cm^−1^ is related to the oscillatory vibration of the CH2 bond. The band appearing at 1033 cm^−1^ is commonly found in non-lignified fibers (NLFs) and is associated with cellulose groups and some carbohydrates. Finally, the band at 770 cm^−1^ is correlated with vibrations of esters and aromatic rings.

Those previous results illustrated in Table 1 and in Figure 2 and Figure 3 indicate a preliminary, possible potential of babassu fibers as reinforcements for polymer composites for technological applications. However, additional results are needed to consolidate this potential for composite reinforcement.

Given the limited information available on babassu fiber, our present study aims to complement the previous results [21] and investigate its chemical and morphological properties through constituent index tests and X-ray diffraction analysis. Additionally, morphological analyses using scanning electron microscopy were conducted.

## 2. Materials and Methods

### 2.1. Materials

The babassu fibers used in this study were obtained in Santana do Maranhão, a city located in the state of Maranhão, Brazil. The process of extracting the fibers from the shell of the babassu coconut involved several steps. First, the shell was broken and exposed to sunlight for two days to facilitate the removal of the fibers. Subsequently, the shell was immersed in water for 14 days to increase the defibrillation of the fibers. After the period of immersion in water, the fibers were removed and showed greater malleability. While the larger diameter fibers could be easily removed, the finer ones required the use of tools such as knives and tweezers, with the necessary precaution to avoid damaging the integrity of the fiber during this process. The comprehensive stages of the babassu fiber extraction process are outlined in detail in the flowchart shown in Figure 4.

### 2.2. Characterization

#### 2.2.1. Diameter Measurement

During the analysis, a significant variation in the diameter of each babassu fiber was observed, necessitating the utilization of the diameter averaging method. The initial step in characterizing the babassu fibers involved evaluating their diameter variation. One hundred fibers were selected for analysis using light microscopy. The equipment employed for the analysis of diametral variation was an optical microscope equipped with a built-in digital camera (Olympus—BX53M, Shinjuku-ku, Tokyo, Japan). The microscope was configured to a 5× amplitude and operated in dark field mode.

The one hundred fibers were measured at three distinct points along their length. At each point, six readings were taken, consisting of three measurements at 0° and three at 90°, resulting in a total of 18 data collection points along the fiber’s length. Optical microscopy played a crucial role in determining the cross-section geometry of the fibers. For each babassu fiber, the average observations at 0° and 90° were considered as the major and minor diameters of the ellipse [22,23,24].

#### 2.2.2. Moisture Content

To determine the moisture content of babassu fiber, a sample weighing approximately 2.0 g should be carefully measured and placed in a designated container. The sample, along with the container, is then placed in an oven set at a controlled temperature of 105 ± 2 °C for 3 h. This process should be repeated until a consistent mass is obtained, indicating no further variation in moisture content. The moisture content is calculated using Equation (1), as outlined in previous studies [25,26,27].
(1)$$MC\%=\frac{\left(MWSA-MDSA\right)}{MWS}\times 100\%$$
where: (MWS) is wet sample mass, (MWSA) is mass of the container/wet sample set, (MDSA) is mass of the container/dry sample set, and (MC) is moisture content.

#### 2.2.3. Determination of Chemical Composition

##### Determination of the Extractive Content

The determination of the extractable content of babassu fiber was performed using the Soxhlet extraction method. Before starting the procedure, it was ensured that the equipment used was properly cleaned and free from grease. A cartridge containing approximately 4000 g of dry matter from the plant was prepared and inserted into the equipment.

Next, a toluene/ethanol mixture in a 1:1 (*v*/*v*) ratio was added to a 500 mL Erlenmeyer flask. The system was refluxed for a period of four to five hours, with a minimum of 24 reflux cycles.

After the extraction time was completed, the solvent used was recovered using a rotary evaporator. The resulting extract was transferred to a pre-cleaned and dried Petri dish, avoiding possible losses (small volumes of fresh solvent were used if necessary). The dish was then placed in an oven at 105 ± 2 °C for 1 h and subsequently cooled in a desiccator. After cooling, the dish was weighed.

This process of heating and weighing was repeated for 30 min until a constant mass was obtained. Equation (2) was used to determine the extractable content [25,26].
(2)$$EC\%=\frac{\left(MSRS-MES\right)}{MS}\times 100\%$$
where (MS) is sample mass, (MES) is mass of the plate/extractives set, (MSRS) is mass of the plate/solvent residue set, and (EC%) is extractives content in percent.

##### Determination of Ash Content

To determine the ash percentage, the following steps were followed. Firstly, 2.0 g of dry babassu fiber were carefully weighed and placed into a pre-calcined crucible, which was heated to 600 °C for 30 min. The crucible containing the fiber was then transferred to a muffle furnace at room temperature. Inside the muffle furnace, a heating ramp of 9.6 °C per minute was applied, gradually increasing the temperature until it reached the target working temperature of 600 °C within 1 h. This temperature was maintained for a duration of three hours to ensure complete combustion of the fiber. Following the three-hour period, the temperature of the furnace was gradually lowered to 200 °C and held constant for one hour. At this point, the crucible sample set was carefully removed from the furnace and placed in a desiccator to cool down and prevent moisture absorption.
(3)$$ASC\%=\frac{\left(MCS-MC\right)}{MS}\times 100\%$$
where: (MS) is mass of the sample, (MC) is mass of the container, (MCS) is mass of the calcined set, and (ASC%) is ash content in percent.

##### Determination of Lignin Content

Lignin determination requires the use of approximately 1.0 g of finely ground sample, which should be devoid of moisture and extractives. The babassu sample should be carefully placed in a grate and mixed with at least 300 mL of 72% (*w*/*w*) sulfuric acid, which has been cooled to a temperature of 10 to 15 °C prior to usage. The mixture should be vigorously stirred with a pestle for 15 min until all visible particles are dissolved. Then, the solution should be transferred to a beaker and allowed to sit undisturbed for 24 h.

Following the previous steps, 306 mL of distilled water was added to the Becker flask to dilute the sulfuric acid concentration to 4%. The content was quantitatively transferred to a 500 mL Erlenmeyer flask and placed on a hot plate. A condenser was connected to the flask and initiated reflux, maintaining the material under reflux for a duration of 4 h.

After completing the aforementioned procedures, a filtration system was set up using a 1000 mL kitassat and a vacuum pump. The precipitate was collected in a funnel and washed until its pH is similar to that of the water used. Subsequently, the funnel was placed in an oven set at 105 ± 2 °C for a period of three hours, allowing it to dry until a constant weight is achieved. Finally, the combined weight of the funnel and the lignin was measured using a highly accurate analytical balance with a precision of 0.0001 g. Equation (4) provides the formulation for determining the lignin content [25,26,28,29,30,31].
(4)$$LC\%=\frac{\left(MFL-MF\right)}{MS}\times 100\%$$
where: (MF) is mass of the clean and dry funnel, (MS) is on mass of the sample, (MFL) is mass of the funnel added to the mass of lignin, after oven drying, and (LC%) is lignin content.

##### Determination of Holocellulose Content

Dried babassu fibers, free from extractives, were subjected to extraction using a 24% potassium hydroxide solution. The fibers were covered with the solution, and the extraction process was carried out at room temperature for a duration of 24 h [25,26,29,30,31].

To proceed further, the mass of the sintered glass funnel should be accurately measured. Subsequently, the funnel was placed on top of a 1000 mL kitassat and connected to a vacuum pump. The contents of the flask were transferred to the funnel, and the precipitate was washed with distilled water until it reached a neutral pH. Afterward, the funnel was positioned in an oven at 105 ± 2 °C and allowed to dry for a minimum of 18 h until a constant weight is achieved. The formulation for determining the holocellulose contents can be found in Equation (5).
(5)$$HC\%=\frac{\left(MFH-MF\right)}{MS}\times 100\%$$
where: (MF) is mass of the clean and dry funnel, (MS) is mass of the sample, (MFH) is mass of the funnel added to the mass of holocellulose, and (HC%) is holocellulose content.

##### Determination of the Alphacellulose Content

The dried holocellulose from the previous test was used, weighed to about 1.0 g, and placed in a crucible of at least 100 mL. A total of 15 mL of a 17.5% sodium hydroxide solution was added, with a wait of two minutes of contact between the solution and the cellulose, and then the material was ground for eight minutes. Then, 40 mL of distilled water was added to the mortar, and the contents were quantitatively transferred to the funnel. The precipitate collected in the funnel should be rinsed until the filtrate has a neutral pH. The funnel should then be dried at 105 ± 2 °C (>18 h), placed in the desiccator, and weighed to constant weight (mass of alphacellulose). Equation (6) shows the method for determining the alphacellulose content [25,26,29,30,31].
(6)$$AC\%=\frac{\left(MFH-MF\right)}{MS}\times 100\%$$
where: (MF) is mass of the clean and dry funnel, (MHS) is mass of the holocellulose sample, (MFA) is mass of the funnel added to the mass of alphacellulose, and (AC%) is alphacellulose content.

### 2.3. X-ray Diffraction (XRD)

To perform the XRD analysis, the babassu fibers were dried in an oven at 75 °C for 24 h. The fibers were cut to a width of 60 mm and mounted in parallel on a monocrystalline silicon substrate, as shown in Figure 5 [32]. The analysis was performed using the Xpert Pro MRD System equipment from Malvern PANalytics (Malvern, UK) with Cobalt Kα radiation (1.789 A), with a scan speed of 4°/min, a power of 40 mA × 40 kV, and scanning from 10° to 60°. An XRD analysis can obtain the diffraction profile of the babassu fiber in natura and thus determine parameters such as the crystallinity index (CI) and microfibril angle (MFA) [33,34].

#### 2.3.1. Crystallinity Index

The method proposed by Segal et al. [35] was used to calculate the crystallinity index (Ic). The value is found from the relationship that uses the intensity of the peak (002) that is considered the crystalline peak and the intensity of the amorphous part (001), according to Equation (7) [35,36,37,38].
(7)$$CI\%=\frac{\left(I_{002}-I_{001}\right)}{I_{002}}\times 100\%$$
where (CI) is crystallinity index, I002 is peak referring to the crystalline part, and I001 is peak referring to the amorphous part.

#### 2.3.2. Microfibril Angle

The microfibril angle (MFA) value was obtained through a series of derivations of the Gauss curve referring to the crystalline plane of the peak (002). To obtain the Gaussian curve, the Origin Pro software version 2023b was used, using some steps to determine the T value. To obtain it, it is necessary to remove the baseline from the diffractogram, and thus, it is possible to determine the Gaussian of the 002 plane [39,40]. The T value is called the angle between the line from the center of the Gaussian peak to the intersection point between the first derivative and the second derivative. To obtain it, it is necessary to use Equation (8):(8)$$MFA=-12.198T^{3}+113.67T^{2}-348.4T+358.09$$

### 2.4. Scanning Electron Microscopy (SEM)

The morphological surface characterization of babassu fibers was performed via scanning electron microscopy (SEM). The model used was a Quanta FEG 250 Fei equipped with a secondary electron detector, operating at an acclimation voltage of 10 kV. To be able to obtain the SEM images of the babassu fiber, the fibers were coated with gold using a Leica ACE600 sputtering machine [41,42,43].

## 3. Results and Discussion

### 3.1. Diameter Variation

The FNLs have some disadvantages, such as their heterogeneity. These characteristics also apply to the diameter, which varies considerably along its length. Additionally, babassu fibers also exhibit significant differences in diameters when comparing different samples [21,22,23]. To analyze this aspect, the frequency distribution method was used to analyze the diameter of 100 babassu fibers, with the aid of an optical instrument. Figure 6 illustrates the optical microscopy of a babassu fiber, revealing a marked heterogeneity throughout its development.

In Figure 4, it is possible to observe the variation of the fiber diameter along its length. This difference in diameter can be explained by the significant variation in the length of the babassu fiber, which sets it apart from other fibers. Several factors can influence this diametral variation, such as the extraction of the fibers from the babassu coconut, fiber drying, water immersion, and equipment handling during analysis [44,45]. The results obtained from the analysis of the diametral variation of the babassu fiber in relation to the diameter are presented in Figure 7.

The structures were discovered within the range of average (0.132 mm) to maximum (0.470 mm) values. The third highest number of fibers was observed within the range of 0.19 mm to 0.252 mm, with a total count of 21 fibers. On the other hand, the lowest distribution frequency was found within the range of 0.132 mm to 0.153 mm, comprising only four fibers. Notably, the fibers with the largest diameter were located in the second range, exhibiting a considerable number of diameters measuring 0.472 mm. However, it is important to note that this lower frequency is attributed to difficulties encountered in unwinding thicker fibers. These challenges arise due to the larger internal tasks involved and the tendency for such fibers to break more easily during the process. Overall, the frequency distribution values suggest an average fiber diameter of 0.27 mm.

### 3.2. Determination of Chemical Composition

The moisture content and chemical composition values for the babassu fibers are shown in Table 2.

The organization of lignocellulosic fibers, comprising cellulose, hemicellulose, and lignin, within the fiber structure is a complex process. An accurate understanding of the composition and arrangement of these components is crucial for effectively utilizing natural fibers in the production of polymer matrix composites.

In this study, the moisture content of babassu fibers was compared with data presented by other researchers, as observed in Table 1. The observed value closely aligns with moisture levels reported for other NLFs. This variation can be attributed to several factors that directly influence the properties of NLFs, such as cultivation site, plant age, and storage conditions [53]. The low moisture content of babassu fiber makes it suitable for application in polymeric matrices for composite production. The hydrophilic nature of fibers with lower moisture content reduces the percentage of water inside the hydrophobic matrix, facilitating a more efficient interaction between the materials [54].

Among the NLFs studied, the babassu fiber exhibits a higher content of extractives, particularly waxes, which may be associated with the presence of waxy dust in the analyzed fiber. Waxy substances in natural fibers generally influence wettability and adhesion properties [55]. As for ash content, the babassu fiber presents a value similar to the other fibers listed. A higher lignin content was observed in the fiber samples analyzed (28.53%), compared to the other data presented. The lignin content in fibers plays a key role in the structure, properties and morphology. Lignin plays the role of an amorphous binder between the fibrils, promoting the connection between cellulose and hemicellulose. It is present in all layers of the fiber cell wall, with higher concentration in the primary and secondary layers [55,56].

The concentration of hemicellulose found in the present research (32.34%) was higher than the other analyzed fibers, which indicates an excellent elasticity property of babassu fiber. Besides being water soluble, the presence of hemicellulose also plays an important role by preventing direct contact between microfibrils [57].

The cellulose content found in babassu fiber was 37.97%, showing to be lower than some other fibers mentioned. However, the cellulose parameter in the fiber still indicates good mechanical properties, since cellulose plays a crucial role in the cell wall strength of plant cells. Therefore, it is directly related to the mechanical performance of fibers [10,58].

### 3.3. XRD Results

Figure 8 presents the diffractogram of babassu fiber, displaying two prominent peaks: one amorphous and one crystalline. These peaks correspond to the planes (101) and (002), with angles of 19.26° and 25.37°, respectively. The (101) peak is attributed to non-cellulosic components like hemicellulose and lignin, while the (002) peak is associated with the crystalline and cellulosic constituent of the natural fiber [59,60,61]. The intensity of the (002) peak is directly linked to the cellulose content, indicating that a higher cellulose content results in a more intense peak [62].

Based on the intensities observed for the (101) and (002) planes, Equation (7) can be applied. The calculated crystallinity index of the babassu fiber was determined to be 81.06%. This value is superior when compared with the value obtained in the work of Rodrigues et al. [63] for babassu. On the other hand, Maniglia et al. [64] obtained a lower value compared to the findings of the current study. Several factors, including cultivation location, plant age, and soil properties, can directly influence the crystallinity value [53].

To determine the microfibril angle (MFA), the diffractogram of babassu fiber in its natural state was utilized, specifically focusing on the (200) plane corresponding to the crystalline structure of the fiber. By employing Origin Pro 2023b Software and incorporating the crystallographic file 00-056-1718, the MFA of babassu fiber was calculated [65]. Upon examination of Figure 6, it is evident that the peak associated with the (002) plane exhibits higher intensity, signifying the characteristic of the crystalline phase (cellulose) and serving as a reference for MFA determination [35,65]. Figure 9 illustrates the methodology employed to derive the T value.

By utilizing the T value obtained from the intersection of the Gaussian curve, the first and second derivatives in Equation (8) yielded a value of 7.64° [66]. This value was found to be lower than that reported for eucalyptus by Hein et al. [67]. However, the value obtained for the guaruman fiber by Reis et al. [65] was similar to the results obtained, as depicted in Table 3.

### 3.4. Microstructural Analysis

SEM images of the babassu fibers are shown in Figure 10.

The SEM image indicates the presence of massive surface porosity and some depression points on the surface of the babassu fiber, which are characteristics found in other natural fibers [68,69]. The existing features in the micrograph of the fiber surface indicate that good adhesion is possible when applied as reinforcements in a polymer matrix composite. At 4000× magnification, at a point of depression, it is possible to observe the presence of roughness and depression, a characteristic that indicates a good anchorage point. This is indicative of the good mechanical properties of babassu fibers [69].

## 4. Conclusions

The analysis of babassu fiber properties reveals promising characteristics, especially in terms of mechanical properties and crystallinity indicators. SEM micrographs demonstrate considerable surface porosity, along with roughness and depression points. These characteristics contribute to good adhesion when the fibers are used as reinforcements in polymer matrix composites. When evaluating the diameter of the fibers, an average of 0.2 mm was obtained, with a range between 0.132 and 0.40 mm. The chemical analysis of the fiber revealed satisfactory parameters for its main constituents, with lignin contents of 28.53%, hemicellulose of 32.34%, and cellulose of 37.97%. These values are higher than those found in some other fibers studied. The diffractogram of the babassu fiber exhibited a pattern characteristic of natural lignocellulosic fibers, while the crystallinity index reached 81.06%, an extremely high value compared to other fibers. In addition, the microfibrillar angle was determined to be 7.64°, indicating good mechanical properties of the fiber. 

## Figures and Tables

**Figure 1 polymers-15-03863-f001:**
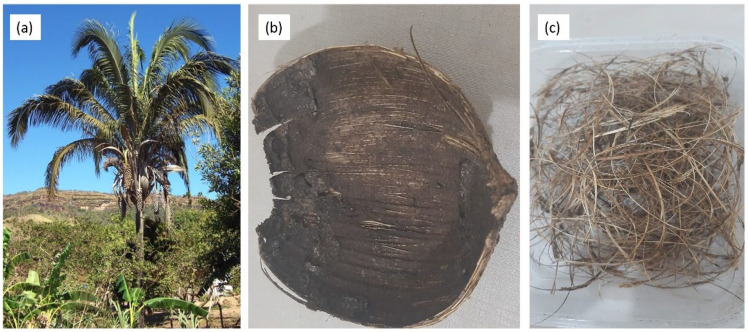
Babassu palm (*Attalea speciosa Mart. ex Spreng.*): (**a**) palm; (**b**) coir; and (**c**) extracted fiber.

**Figure 2 polymers-15-03863-f002:**
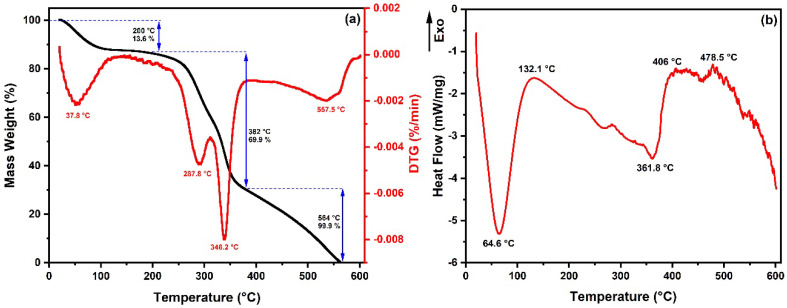
Results of thermal analysis of babassu fibers: (**a**) TG/DTG; (**b**) DSC.

**Figure 3 polymers-15-03863-f003:**
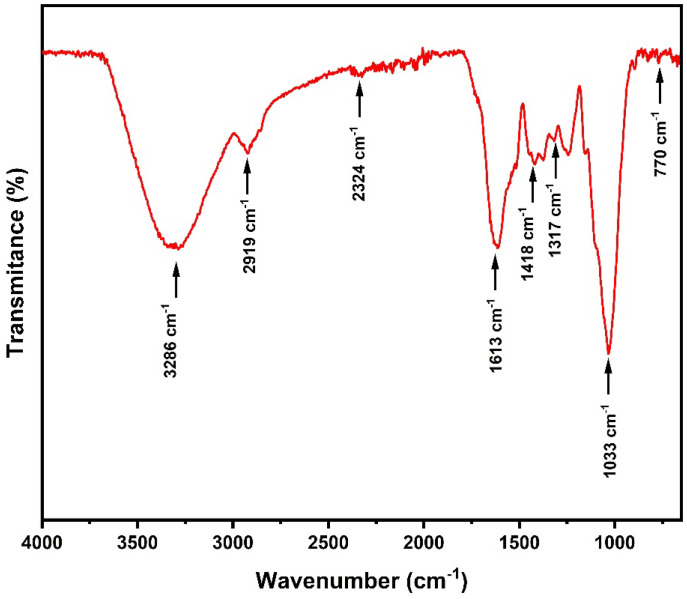
FTIR spectrum for babassu fibers.

**Figure 4 polymers-15-03863-f004:**
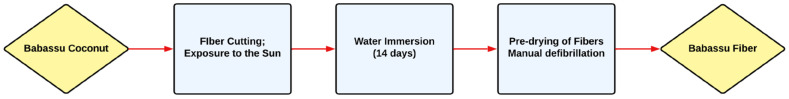
Flowchart indicating the steps to obtain babassu fibers.

**Figure 5 polymers-15-03863-f005:**
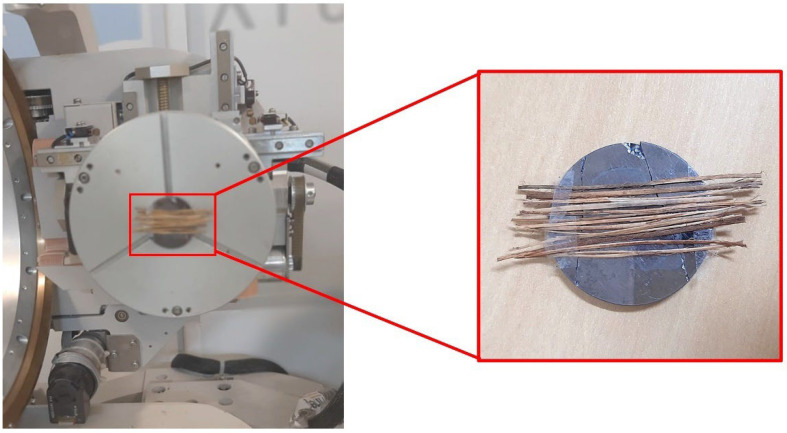
Positioning of babassu fibers on the monocrystalline silicon substrate and placement of the fiber in the diffractometer for analysis.

**Figure 6 polymers-15-03863-f006:**
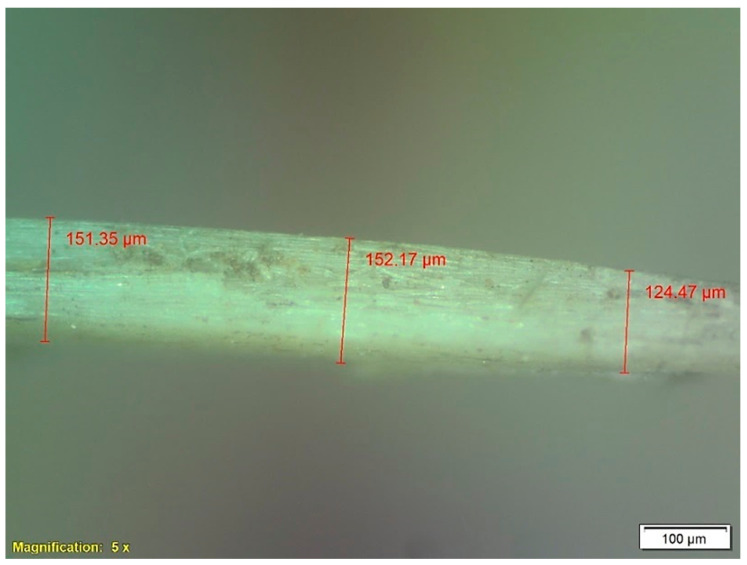
Diameter measurements from the cross-section of the babassu fiber. Magnification of 5×.

**Figure 7 polymers-15-03863-f007:**
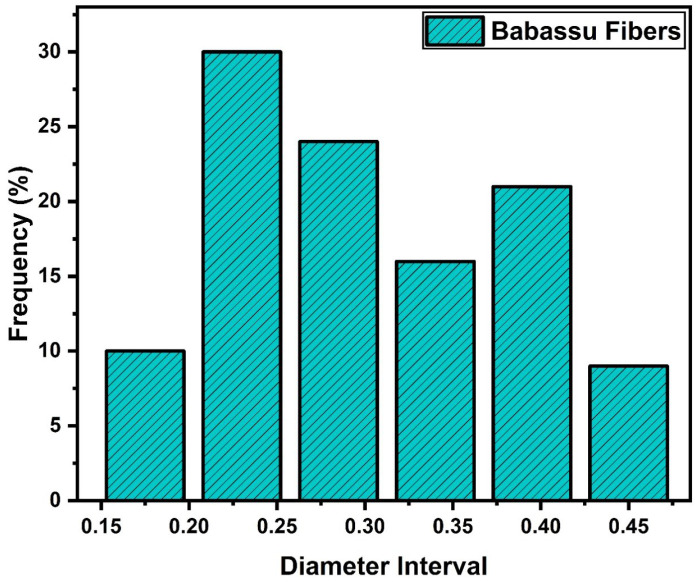
Frequency histogram by diameter interval of babassu fibers.

**Figure 8 polymers-15-03863-f008:**
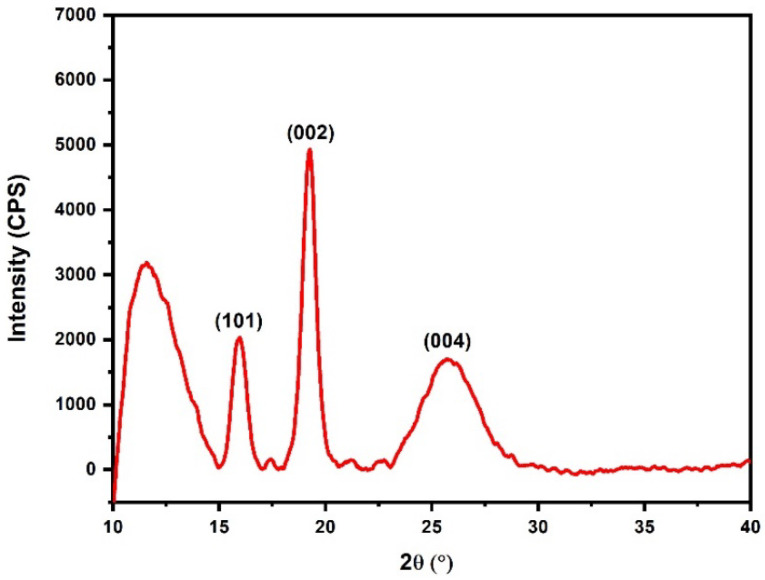
Diffractogram of babassu fiber.

**Figure 9 polymers-15-03863-f009:**
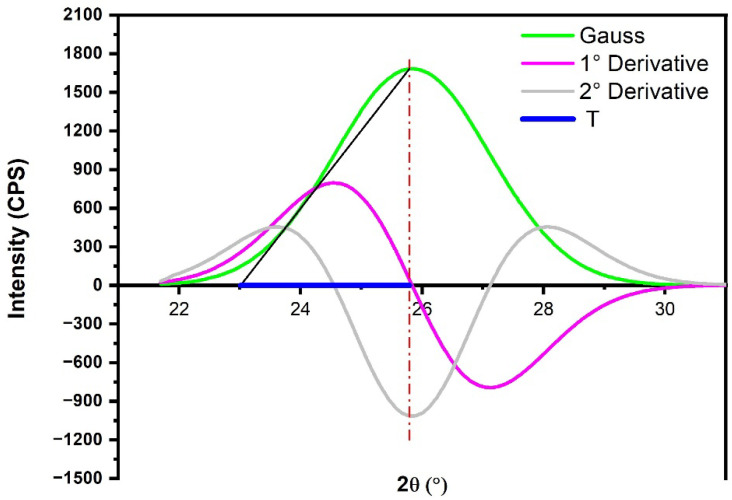
Methodology for determining the microfibril angle of babassu fiber.

**Figure 10 polymers-15-03863-f010:**
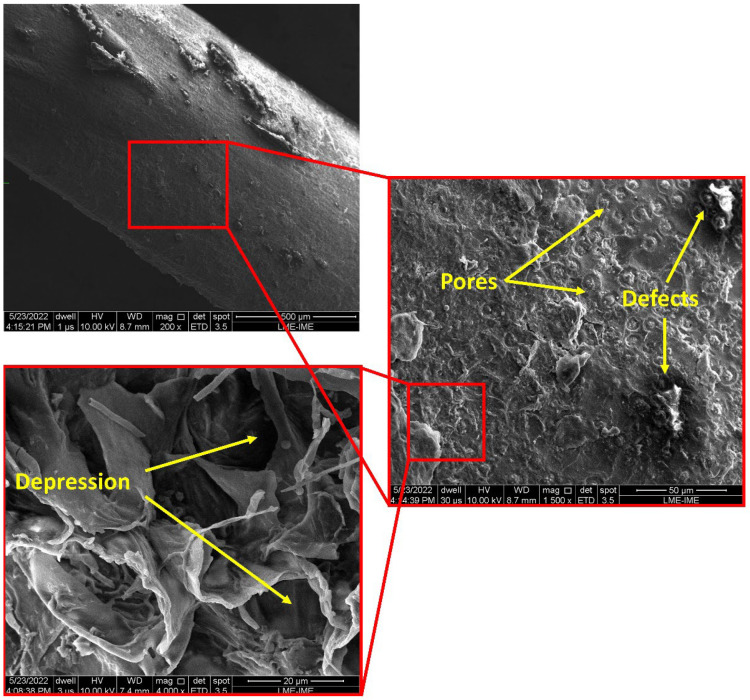
SEM micrograph of the babassu fiber surface indicating fiber morphology at different magnifications: 100, 1500, and 4000×.

**Table 1 polymers-15-03863-t001:** Babassu fiber density and tensile results as a function of fiber diameter variation. Reproduced with permission from [21].

Diameter Intervals (mm)	Average Density (g/cm^3^)	Ultimate Tensile Strength (MPa)	Elongation (%)	Elastic Modulus (GPa)
0.18–0.23	0.79 ± 0.03	100.76 ± 10.18	1.59 ± 0.16	6.33 ± 0.64
0.23–0.29	0.70 ± 0.04	76.41 ± 7.72	2.98 ± 0.30	2.56 ± 0.26
0.29–0.34	0.66 ± 0.03	47.61 ± 5.32	2.05 ± 0.20	2.32 ± 0.23
0.34–0.40	0.53 ± 0.04	31.18 ± 3.15	1.18 ± 0.11	2.87 ± 0.29
0.40–0.45	0.42 ± 0.03	24.97 ± 2.52	1.94 ± 0.20	1.94 ± 0.13
0.45–0.47	0.27 ± 0.01	17.96 ± 0.61	1.56 ± 0.15	1.15 ± 0.11

**Table 2 polymers-15-03863-t002:** Chemical composition of babassu fiber compared to other fibers.

Fiber	MC%	EC%	ASC%	LC%	HC%	HEC% *	AC%	Ref.
Babassu	7.05	15.72	2.43	28.53	70.31	32.34	37.97	p.a *
Babassu	-	8.50	-	21.9	74.50	8.90	65.50	[46]
Jute	12.6	0.50	-	12.00	74.60	13.60	61.00	[47]
Mudar	-	-	2.50	18.00	76.00	-	57.00	[48]
Kenaf	-	-	2.20–6.00	14.00–17.00	76.00–77.00	-	45.00–46.00	[49]
Ficus	9.33	-	3.96	10.13	-	13.86	55.38	[50]
Cactus	5.80	-	-	13.70	-	8.20	67.40	[51]
Soapbark	12.0	-	5.00	18.00	-	20.00	37.00	[52]

p.a *—present article; HEC *—hemicellulose content.

**Table 3 polymers-15-03863-t003:** Comparison of X-ray diffraction properties of babassu fibers with other fibers.

XRD Properties	Babassu (P.A *)	Babassu [63]	Guaruman [65]	Eucalyptus [67]	Seven-Islands-Sedge [40]
MFA%	7.64	-	7.80	13.90	7.36
CI%	81.06	15.00–45.00	67.00	-	62.47

P.A *—present article.

## Data Availability

The data presented in this study are available upon request from the corresponding author.

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
