# Peer review of "Babassu Coconut Fibers: Investigation of Chemical and Surface Properties (Attalea speciosa.)"

_polymers, 2023, doi:10.3390/polym15193863_

Round 1

Reviewer 1 Report

This manuscript investigated the chemical and morphological properties of babassu fibers by X-ray diffraction, and discussed chemical composition including the crystallinity index and the microfibril angle of babassu fibers compared to other fibers. As well as pointed out the babassu fibers could be used as reinforcement in composites.

Unfortunately, there is no novel academic point to address to readers, and the experimental analysis was too simple. The authors should do in-depth characterization of chemical properties of babassu fibers. For examples, elemental analysis, thermogravimetric analysis and mechanical properties. Therefore, I cannot recommend publication of this manuscript in its present form.

Moderate editing of English language required

Author Response

I would like to express my sincere thanks for your comments and the suggested changes to the article. Each of the proposed changes has been carefully addressed and is explained in detail in the following document.

Thank you again for your time and expertise.

Reviewer 2 Report

The experimental article "Babassu coconut fibers: Investigation of chemical and surface properties (Attalea speciosa.)" is devoted to an actual problem: the study of lignocellulosic biomass with a view to its further application. The article corresponds to the profile of the publication Polymers, however, it has a number of shortcomings, after the elimination of which the article can be published.

Notes:

1. Paragraph 2.1 is written incorrectly. The authors begin to describe the technology for extracting babassu fiber from coconut shells, then give up and simply write: "The steps involved in the ex-81 traction of babassu fiber are described in the flowchart of Figure 2." Then it would be necessary not to start describing the fiber extraction technology at all, and simply write that it is presented in the diagram. Or continue the description. And this option is preferable, since it is interesting how laborious the technology of fiber extraction is. And it would be good to give a forecast on the possibility of introducing this technology into production.

2. Clause 2.2.3.3. and 2.2.3.5. why were such methods of estimating the chemical composition chosen? And not traditional? For example, Klasson's lignin?

Author Response

(The authors gave the same response as above.)

Round 2

Reviewer 1 Report

The authors have revised the manuscript and added the TG-DSC curves, FTIR spectrum of babassu fibers. However, there is no corresponding description and discussion of these data. Meanwhile, I still suggest that the author should further refine the objectives and research highlights of the work. Therefore, I cannot recommend publication of this manuscript in its present form.

Moderate editing of English language required.

Author Response

The authors have revised the manuscript and added the TG-DSC curves, FTIR spectrum of babassu fibers. However, there is no corresponding description and discussion of these data. Meanwhile, I still suggest that the author should further refine the objectives and research highlights of the work. Therefore, I cannot recommend publication of this manuscript in its present form.

Response:

The reviewer's observation proved to be fundamental, as he noticed that the graphs were presented in the previous article without a proper explanation of the results obtained. We are grateful for the suggestion to include explanatory text for the TGA-DTG, DSC and FTIR graphs. In the new revision, we have implemented all the corrections and suggested texts for the graphs recommended by the reviewer.

Round 3

Reviewer 1 Report

accept

Minor editing